# Mangiferin Affects Melanin Synthesis by an Influence on Tyrosinase: Inhibition, Mechanism of Action and Molecular Docking Studies

**DOI:** 10.3390/antiox12051016

**Published:** 2023-04-28

**Authors:** Anna Hering, Justyna Stefanowicz-Hajduk, Szymon Dziomba, Rafal Halasa, Radoslaw Krzemieniecki, Subrahmanyam Sappati, Maciej Baginski, Jadwiga Renata Ochocka

**Affiliations:** 1Department of Biology and Pharmaceutical Botany, Medical University of Gdansk, 80-210 Gdansk, Poland; 2Department of Toxicology, Medical University of Gdansk, 80-210 Gdansk, Poland; 3Department of Pharmaceutical Microbiology, Medical University of Gdansk, 80-210 Gdansk, Poland; 4Department of Pharmaceutical Technology and Biochemistry, Faculty of Chemistry, Gdansk University of Technology, 80-233 Gdansk, Poland

**Keywords:** xanthone, antioxidant, enzyme inhibition, L-DOPA, food browning, pigment, complex formation, active center, peripheral site

## Abstract

Mangiferin is a strong antioxidant that presents a wide range of biological activities. The aim of this study was to evaluate, for the first time, the influence of mangiferin on tyrosinase, an enzyme responsible for melanin synthesis and the unwanted browning process of food. The research included both the kinetics and molecular interactions between tyrosinase and mangiferin. The research proved that mangiferin inhibits tyrosinase activity in a dose-dependent manner with IC_50_ 290 +/− 6.04 µM, which was found comparable with the standard kojic acid (IC_50_ 217.45 +/− 2.54 µM). The mechanism of inhibition was described as mixed inhibition. The interaction between tyrosinase enzyme and mangiferin was confirmed with capillary electrophoresis (CE). The analysis indicated the formation of two main, and four less significant complexes. These results have also been supported by the molecular docking studies. It was indicated that mangiferin binds to tyrosinase, similarly to L-DOPA molecule, both in the active center and peripheral site. As it was presented in molecular docking studies, mangiferin and L-DOPA molecules can interact in a similar way with surrounding amino acid residues of tyrosinase. Additionally, hydroxyl groups of mangiferin may interact with amino acids on the tyrosinase external surface causing non-specific interaction.

## 1. Introduction

Polyphenols are a broad group of natural compounds with antioxidant properties that can be used as a protective agent against UV radiation, smog, or smoking cigarettes that may lead to lifestyle-related diseases. Polyphenols obtained mostly from well-known plant materials are used as nutrients, food additives, and cosmetic ingredients for protecting macromolecules from decomposition and the aging process [1,2,3,4].

Mangiferin, a natural C-glucosylxanthone (Figure 1), occurs abundantly in nature. The main source of this compound is mango (*Mangifera indica* L., Anacardiaceae). A large amount of mangiferin was also detected in the plant families of Liliaceae, Fabaceae, Hypericaceae, and many other plants [5,6,7,8,9]. As a remarkably reactive compound, the isolation of mangiferin and its bioactivity have been studied by several research institutions [10,11,12]. Because of its specific structure, mangiferin is a strong antioxidant and functions as a protective agent, thus it is capable of protecting organs and skin from the harmful effects of oxidative stress and UV radiation [11,13,14,15,16,17]. In the presence of reactive oxygen species (ROS), mangiferin modulates the activity of enzymes and/or the expression of genes that play a critical role in autoimmune diseases, activation of apoptosis, viral replication, and tumorgenesis [10,11,13,17,18,19,20,21]. Mangiferin inhibits the inflammation process as well as bacterial and fungal growth, which are responsible for cell destruction and activation of tyrosinase [19,22,23,24]. 

Melanin is a large pigment composed of polyphenolic compounds [25]. Melanin performs many biological functions and creates a link between the organism and the external environment. The photoprotective property of melanin seems to be its most important function as thermoregulation, camouflage, and expression of sexual activity are essential to survival. In animals, melanin is responsible for the color of skin, hair, and eyes. The melanin pigments absorb UV light and protect the deep layers of the skin from the harmful effects of sunlight [25,26,27,28,29,30]. 

The most important enzyme involved in melanin formation is tyrosinase. Belonging to the class of oxidases, tyrosinase catalyzes not only the ortho-hydroxylation of monophenols and diphenols, known as monophenol and diphenol cycles respectively but also the oxidation of L-DOPA into reactive o-quinones. The active site of tyrosinase is composed of two copper ions surrounded by histidine residues, and the enzyme occurs in three forms: deoxy-tyrosinase, oxy-tyrosinase, and met-tyrosinase [31]. The multifunctional enzyme exhibits broad substrate specificity, with a high affinity to L-isomers [32]. 

In plants, two forms of tyrosinase have been described: membrane-bound and soluble. The gene encoding tyrosinase is located in the nucleus, while the translation process occurs in the cytoplasm. The pro-enzyme is transported to the chloroplast where the active form is produced. The substrate is located in the vacuole [33]. The reaction occurs after cutting or crushing the plant material, which leads to the destruction of cell structure, and tyrosinase reacts with its substrate in the presence of oxygen. At the appropriate temperature and pH, the browning process occurs rapidly and leads to the spoilage of agricultural products during storage [26,32]. As a consequence, the flavor, texture, and nutritional quality of the food are changed. Less attractive crops, beverages, and seafood generate significant financial losses to the food industry. Therefore, researchers are attempting to find an effective reducing agent and/or inhibitor of tyrosinase activity that is economical and nutritionally safe [34].

To maintain an attractive food appearance, the use of antioxidants and copper-chelating compounds as tyrosinase inhibitors has been analysed [35,36,37]. The chemical compounds used to inhibit the browning process of food products must not only be safe for humans but they should also be relatively cheap to produce. Of many chemicals used for this purpose in the food market, polyphenols obtained from plants are the largest potential group of compounds applied as protective agents [35].

The present study aimed to evaluate the direct influence of mangiferin—the xanthone with many biological activities—on tyrosinase activity, including the investigation of kinetics and molecular aspects of the interaction between them. According to the currently available databases, this type of study has not yet been conducted.

## 2. Materials and Methods

### 2.1. Materials

Mangiferin, kojic acid, tyrosinase (tyrosinase from mushroom), 3-(3,4-Dihydroxyphenyl)-L-alanine (L-DOPA), phosphate buffer (0.175 mM, pH 6.8), phosphoric acid were sourced from Sigma Chemical Co., (St. Louis, MO, USA), and TRIS-HCl, HCl (77 mM), NaCl, CaCl_2_, acetate buffer (acetic acid 79 mM, sodium acetate 24 mM, pH 3.75) for enzymatic assay, and acetate buffer 0.3 M, pH 3.6 for FRAP assay were sourced from Avantor Performance Materials Poland S. A. FeCl_3_ × 6H_2_O, HPLC-grade methanol, and ethanol were purchased from P.O.Ch. (Gliwice, Poland).

### 2.2. Tyrosinase Inhibition Assay

The activity of mushroom tyrosinase (EC 1.14.18.1) during the oxidation of L-DOPA was estimated by the spectrophotometric method with some modifications [38,39]. The dopachrome formation was monitored continuously at 20 s intervals for 20 min (Epoch BioTek System, Agilent Technologies, Inc., Santa Clara, CA 95051, USA). Room temperature was found to be adequate to analyse tyrosinase activity. To reduce the volume of the reaction mixture, 96-well plates (72636 Frickenhausen, Germany, Greiner Bio-One GmbH) were used.
L-DOPA + 1/2 O_2_ → Dopachrome

The reaction mixture comprised phosphate buffer (0.175 mM, pH 6.8), 20 μL of tyrosinase (120 U), and different concentrations of mangiferin (in phosphate buffer: 0–1000 µg/mL). The reaction mixtures were pre-incubated with mangiferin for 15 min. The addition of L-DOPA (10 mM) started the reaction. Product formation changes were recorded spectrophotometrically at λ = 475 nm. In each analysis, blank probes were prepared, and the results were compared with the standard. The dose-dependent course of inhibition was analysed. 

Kojic acid was used as a standard, for the tyrosinase inhibition assay. 

For the inhibition process conducted for mangiferin and tyrosinase, the reversibility and the type of the inhibition process were investigated. The K_m_, V_max_, and IC_50_ values were calculated with the program GraFit v.7.0 (East Grinstead, West Sussex RH19 3AU, UK, Erithacus Software). 

The kinetic parameters V_o_, K_m_, and V_max_ were calculated according to Michaelis–Menten and Lineweaver–Burk plots and analysed using the program GraFit v.7.0 (Erithacus Software) and Microsoft Excel [13]. 

The Lineweaver–Burk plots were constructed and analysed to estimate the type of inhibition process between mangiferin (0, 0.71, 0.95, and 1.42 mM) and tyrosinase.

The plots of tyrosinase velocity depending on tyrosinase concentrations in the presence of different concentrations of mangiferin (0, 0.71, 0.95, and 1.42 mM) were drawn to estimate the reversibility of mangiferin–tyrosinase interaction.

### 2.3. Capillary Electrophoresis (CE)

The CE experiments were performed with the P/ACE MDQ plus system (Sciex, Framingham, MA, USA), equipped with a PDA detector. An uncoated fused silica capillary (50 µm i.d. × 60.2 cm) was purchased from Polymicro Technologies (West Yorkshire, UK). The capillary and samples were kept in a thermostat at 25 °C. The samples were injected from the short end (10.2 cm of effective length) by using 3.45 kPa pressure. Capillary rinsing was performed at 172.4 kPa. Background electrolyte (BGE) was composed of 50 mM phosphate buffer (pH 6.8).

At the beginning of each working day, the capillary was subsequently rinsed with 0.1 M NaOH solution, water, and BGE (10 min each).

Before each run, the capillary was conditioned with 0.1 M NaOH solution (2 min), water (1 min), and BGE (5 min) followed by dipping the capillary in water to rinse its outer walls. Then, the sample injection was performed (5 s, 3.45 kPa). A short BGE plug was introduced after the sample injection (5 s, 3.45 kPa). The negative polarity mode at 10 kV was applied for 6 min to separate the analytes. The separation process was monitored at 238 nm.

For CE experiments, the stock solution of mangiferin was prepared daily in 20% aqueous DMSO solution at the concentration of 2 mg mL^−1^ and was diluted accordingly with 50 mM phosphate buffer (pH 6.8).

### 2.4. Molecular Docking: Methods and Materials

The protein structure of tyrosinase was obtained from the Protein Database (https://www.rcsb.org (accessed on 1 April 2023)) [40], with entry ID 4P6S. Docking studies were performed with the Molecular Operating Environment (MOE; © Chemical Computing Group ULC) software. The PDB file was locally stored for further refinement, keeping only one homodimer. The protein structure was prepared, including the solvent particles from the crystal structure as well as two Zn^2+^ ions, and the recognized ligands binding site (L-DOPA) was left empty. For the step where the “QuickPrep” option was selected and consisted of structure conservation and neutralization, the amino acids were, if applicable, allowed to flip, and the proper protonation of molecules was set. During the protein refinement, the nitrogen atom of histidine in ε2 position remained unprotonated, while the δ1 nitrogen atom (except H231) was protonated. Water molecules that were further than 4.5 Å were removed. The receptor and solvent were set to tether with a strength of 10 (measured as force constant). Atoms, which were further than 8 Å from the ligand cavity, were fixed. Finally, the RMS gradient of 0.1 kcal/mol/Å^2^ was set for minimization refinement with the MOE software using AMBER10 and EHT (Extended Huckel Theory) as force field parameters. Two binding centers were selected, where L-DOPA was originally present in the X-ray structure. The placement method used in the docking was “alpha triangle” for the active center and a “proxy triangle” for the peripheral site, with semi-rigid refinement, and the ligands were capable of changing the conformation to best fit the cavity. The protein was rigid during the docking. A proper site in the receptor was defined, which for the active site was a selection of eight amino acid residues (H42, H60, H69, H204, H208, G216, V217, and H231) and for the peripheral site (N10, L12, H13, E93, and T96), by setting an electron density, was based on the original crystal structure of L-DOPA and proper placement and refinement method. The process of adding mangiferin to MOE included a 2D structure import into the software’s database. Next, by using MOE’s “Wash” option, the structure of the molecule was optimized, hydrogen atoms were explicitly added, a proper 3D structure was generated, and the mangiferin was neutralized at pH 7. 

The forcefield was set to AMBER10 for the protein, EHT for the ligands, and zinc ions were parameterized with OPLS-AA for the purpose of the docking process. The docking procedure consisted of generating 30 different poses of which 5 best poses were saved in the output file, independently for the active and the peripheral site. Ligand refinement included minimization, explicit addition of hydrogen atoms, conversion to a 3D structure with reasonable bond lengths, and neutral protonation of the molecule at pH 7.

The electron density was set after the correction of the structure, including the original ligand of the 4P6S protein. The free energy of binding was calculated with the London dG scoring function and this value was refined with the GBVI/WSA dG scoring function.

### 2.5. Statistical Analysis

The statistical data of kinetic studies were analysed from three independent analyses (*n* = 9) using the STATISTICA 12.0 software package (StatSoft Inc., Tulsa, OK, USA). All data are expressed as mean values ± standard deviation (SD). For comparison studies, one-way ANOVA with post hoc Tukey’s test was performed. The statistical significance was set at *p* < 0.05.

## 3. Results

### 3.1. Influence of Mangiferin on Tyrosinase Activity

The influence of mangiferin (0–2.5 mM concentrations) on tyrosinase activity was investigated using L-DOPA as the substrate. The results revealed that tyrosinase activity was inactivated by mangiferin in a dose-dependent manner (Figure 2). A rapid decrease in tyrosinase activity (by approximately 40%) was achieved with 0.237 mM mangiferin concentration. Surprisingly, a constant increase in inhibitor concentration did not significantly affect tyrosinase activity. Inhibitor concentrations of higher than 1 mM did not significantly decrease the enzyme activity. The concentration of mangiferin that led to a 50% loss of tyrosinase activity (IC_50_) was 0.290 ± 0.006 mM, and the IC_50_ value of the standard kojic acid was 0.217 ± 0.002 mM.

The type and reversibility of tyrosinase inhibition were determined with the increasing concentration of mangiferin (0, 0.71, 0.95, and 1.42 mM; Figure 3). The kinetic parameters indicated that with the increase in mangiferin concentration, V_max_ decreased and K_m_ slightly increased. The intersection of the curves is outside the X and Y axes, which indicates a mixed type of inhibition. Thus, mangiferin showed affinity to the free enzyme and the enzyme-substrate complex.

To determine the reversibility of the enzyme-inhibitor binding reaction, plots of the tyrosinase activity against the enzyme concentration were drawn in the presence of different amounts of mangiferin (Figure 4). The plots showed straight lines passing through the origin. As the mangiferin concentration increases, the slope of the lines decreases. Thus, the results confirmed that mangiferin reversibly inhibits tyrosinase activity.

### 3.2. Results of Capillary Electrophoresis

The CE analysis of the mangiferin solution revealed the presence of an impurity (Imp) showing similar electrophoretic mobility and the UV spectrum as the main peak (black trace; Figure 5). The purity of the substance used in the experiments was determined to be 94.2% ± 0.1%.

The addition of tyrosinase decreased the signals of mangiferin and its’ impurity signals, thus indicating interactions. The decrease in the corrected area and height was proportional to the amount of added enzyme. Consequently, additional signals also appeared (numbered 1–6 in Figure 5). The UV spectra of these signals were similar to that of mangiferin. Furthermore, as shown in Figure 5, the total area of anionic species detected in certain samples was constant (CV < 2.2%). The various electrophoretic mobilities indicate that these complexes exhibited diverse charges and/or hydrodynamic sizes [41]. While the formation of a complex with a small molecule is considered to mainly affect the charge of the protein [42], the presence of numerous binding sites in the tyrosinase structure might explain this phenomenon. Under these conditions, the saturation of centers showing the highest affinity to the ligand will be observed. This situation is observed in Figure 5 where the domination of a single and double-ligated form is noticeable (peaks 1 and 2, respectively). Moreover, the complexes showing the highest electrophoretic mobility (the largest number of bound ligands) had the lowest intensity.

It should be emphasized that these studies were conducted in vitro without the addition of a substrate that could change the effect of interactions, and thus the resulting complexes were formed. The experiment indicated that with the increase in enzyme amount, almost all mangiferin molecules interact with tyrosinase in different positions, resulting in the appearance of peaks 1–6. The highest amount of mangiferin was incorporated into tyrosinase with the formation of peaks 1 and 2. Peak 3 is also noteworthy, while the remaining peaks 4–6 are probably of marginal importance in the mangiferin-tyrosinase interactions.

### 3.3. Molecular Docking Studies

To assess whether mangiferin can interact with tyrosinase in two potential binding sites occupied by L-DOPA, docking studies were conducted using the molecular modeling approach. The results confirmed that both active and peripheral sites can be considered binding sites. In the active site, the major interactions between the receptor and the ligand were arene-arene interactions with His208 and arene-H with Val218. This interaction is observed in L-DOPA crystal structure and the docked one, as well as in mangiferin (Figure 6A, Figure 7A and Figure 8A). A salt bridge was observed between Glu158 and the hydroxyl group of glucose. Furthermore, an H-arene (CH-π interaction) interaction was observed with Arg209 wherein the xanthone ring was directly bound to the glucosyl group (Figure 8A). In the peripheral site, a salt bridge connected Arg165 and the glucosyl ring, and multiple non-covalent interactions were observed (Figure 8B). In the peripheral site, L-DOPA is originally bound to His13 through arene-arene interaction as well as with Glu93, thus creating a salt bridge, while the docked L-DOPA showed an interaction only with Glu93 through a salt bridge (Figure 6B and Figure 7B). Docking with mangiferin showed interaction creating an H-aryl interaction through the water molecules, while with His13 is possible to observe ligand-receptor exposure (Figure 8B). The docking scores were more favorable for the active site than the peripheral. The binding energy was similar for the L-DOPA (average score: −6.77 kcal/mol) and mangiferin (average score: −6.84 kcal/mol) in the active center, while in the peripheral site the energy was more advantageous for mangiferin (average score: −4.59 kcal/mol) than for L-DOPA (average score: −3.90 kcal/mol).

On the basis of these results, we propose that both sites, namely, the active center and the peripheral site, can be occupied by mangiferin similarly to the L-DOPA molecule. Furthermore, both molecules can interact identically with the surrounding amino acid residues (Figure 6, Figure 7 and Figure 8). 

## 4. Discussion

Our present study elucidated the interaction between mangiferin and the enzyme tyrosinase. The analysis was based on the data obtained for IC_50_, value, and kinetics of the interaction, i.e., the type of inhibition and its reversibility. CE was used to confirm the formation of the enzyme-mangiferin complex and detailed molecular interactions were presented by molecular docking studies.

The analyses of enzymatic activity demonstrated for the first time the mechanism and ability of mangiferin to inhibit tyrosinase, an enzyme responsible for the first two steps of melanogenesis. This xanthone is capable of inhibiting tyrosinase reversibly in a dose-dependent manner similar to the strong standard kojic acid. The inhibition process occurs immediately after the contact of mangiferin with tyrosinase. In consequence, enzyme-mangiferin complexes are formed at both active and peripheral sites (typical for L-DOPA binding), which suggests that mangiferin can act also on the external surface of the tyrosinase. However, it is highly probable that these surface interactions do not affect the activity of tyrosinase itself.

Our analysis revealed, that without the substrate, mangiferin interacts with tyrosinase with a stronger affinity to the active center than to the peripheral area (peaks number 1 and 2 in Figure 5), and probably toward both active and peripheral areas on one tyrosinase macromolecule (peak number 3 in Figure 5). Therefore, we can expect that three different complexes can be formed as follows: tyrosinase-mangiferin at the active site; tyrosinase-mangiferin in the peripheral site; tyrosinase and mangiferin present simultaneously at both the active and peripheral site. Additionally, possible interactions between the hydroxyl groups of mangiferin and the polar groups of amino acids at the protein surface can cause the formation of additional small peaks 4–6 (Figure 5). Kinetic experiments, conducted in the presence of the substrate, revealed that the interaction between mangiferin and tyrosinase is reversible. The mangiferin competes with the substrate at binding areas both at the active and peripheral sites. It should be noted that mangiferin could not completely inhibit the catalytic capacity of tyrosinase. It might suggest that during the reaction, mangiferin is involved more in redox reactions (through the catechol group in the mangiferin skeleton) with quinones generated by tyrosinase activity, than in enzyme inhibition. The other assumption is that the affinity of mangiferin to amino acids on the peripheral site and the surface can be similar or stronger than mangiferin affinity to the active site, therefore the capability of mangiferin molecules to interact with the active site is limited. Mangiferin probably has a similar affinity to all possible sites (active site, peripheral site, and the surface), and after following the initial stage of inhibition, mangiferin no longer competes with the substrate and is unable to completely inhibit the enzyme activity. Those two possibilities can explain the phenomenon of the unexpected slowdown of tyrosinase inhibition by mangiferin, immediately after reaching the IC_50_ value. Under the present research conditions, a further constant increase in mangiferin concentration during the inhibition process did not alter the remaining tyrosinase activity (25%). The obtained results confirmed that mangiferin at the initial stage of interaction binds to the active site Figure 6A. This was also confirmed by the rapid process of inhibition of the enzyme activity (Figure 2) as well as the formation of the first large peaks in the CE analysis (Figure 5). 

In the present study, we did not assess the quantitative affinity of mangiferin to the enzyme-substrate complex and the process of its inactivation. This aspect requires further analysis. However, according to our results, mangiferin exhibits a mixed type of inhibition by forming several types of complexes with the enzyme. Mangiferin can interact with macromolecules such as tyrosinase by interacting with their active center as well as with the amino acids on the peripheral site and external surface. Because of these interactions, additional complexes are formed, which may limit mangiferin's ability to inhibit enzyme activity completely.

According to our previous research [43], the apoptotic activity of the polyphenol hesperidin is enhanced when accompanied with the antioxidant mangiferin. These findings suggest that the combination of mangiferin and a compound with strong antioxidant properties that are capable to reduce quinones can enhance the inhibition of tyrosinase by mangiferin. First, tyrosinase inhibition might be achieved with the use of lower mangiferin concentration; subsequently, the products generated by tyrosinase could be inactivated before the cascade of melanin synthesis begins. Several types of tyrosinases have been identified as they are widespread in nature and perform various functions. Tyrosinases are present in bacteria, mushrooms, plants, and animals. In many biological processes, the activity of tyrosinase is essential and corresponds with its protective functions. Nevertheless, excessive activity of this enzyme leads to unwanted pigmentation changes and unwanted browning of food. 

In fruits, vegetables, mushrooms, and crustaceans, tyrosinase is one of the main factors of the browning process, which causes a change in nutritional quality and color and makes the product less attractive [35]. Mangiferin could be used as a ripening retardant on the basis of its properties of tyrosinase inhibition and inactivation of the reactive quinones generated as a consequence of L-DOPA destruction. According to the fact that mangiferin has poor intestinal absorption [44], it could be used as a suitable compound for preserving fresh vegetables and fruits from browning after cutting or crushing the plant material. Moreover, small doses of mangiferin absorbed by the intestine can have positive nutritional effects [11,16], and mangiferin can protect tissues and organs from toxins and limit allergic reactions [5,14,15,16,17]. Although mangoes are the main source of mangiferin, it is also found in other so-called ‘waste products’, such as kernels and leaves [10]. Mango kernel extract and its main constituents were tested for antityrosinase activity and showed good results [8]; however, mangiferin by itself was not considered in the previous study. 

High activity of tyrosinase in human skin leads to local hyperpigmentation [25,26,28]. Mangiferin does not show dermal toxic effects [16]. Moreover, our previous analysis revealed that after application on the skin surface, mangiferin can pass through the stratum corneum and penetrate the dermis and epidermis layers [13,45]. On the basis of these findings and following additional, comprehensive in vitro and in vivo analysis with human tyrosinase and appropriate cell cultures, mangiferin could be considered a candidate for further research in the field of skin pathological conditions as well as for application in the cosmetic industry to maintain regular skin color.

## 5. Conclusions

The presented analysis confirmed the ability of mangiferin to inhibit L-DOPA oxidation caused by tyrosinase. Therefore, mangiferin can be used to inhibit the unwanted overproduction of melanin and be utilized to inhibit the browning process in the food industry.

## Figures and Tables

**Figure 1 antioxidants-12-01016-f001:**
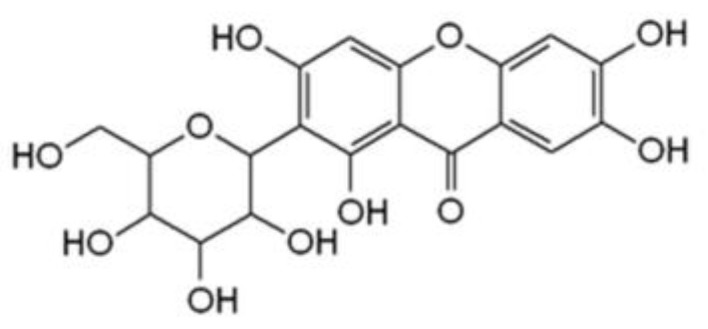
Mangiferin chemical structure (C_19_H_18_O_11_).

**Figure 2 antioxidants-12-01016-f002:**
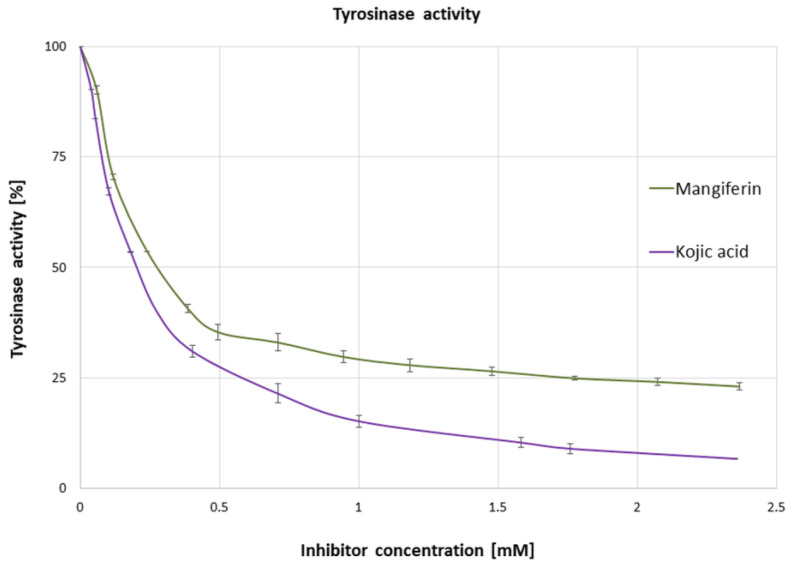
Changes in the activity of tyrosinase (%) in the presence of increasing concentrations of mangiferin and kojic acid (mM). The results were obtained from three independent experiments, in three replicates (n = 9). Error bars represent standard deviations.

**Figure 3 antioxidants-12-01016-f003:**
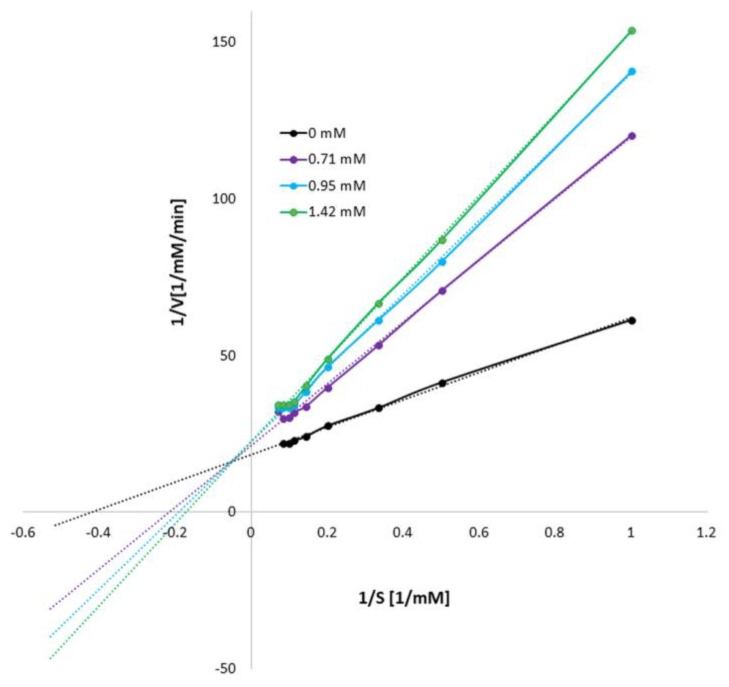
Lineweaver-Burk plot of L−DOPA oxidation with tyrosinase in the presence of mangiferin (0 mM, 0.71 mM, 0.95 mM, and 1.42 mM).

**Figure 4 antioxidants-12-01016-f004:**
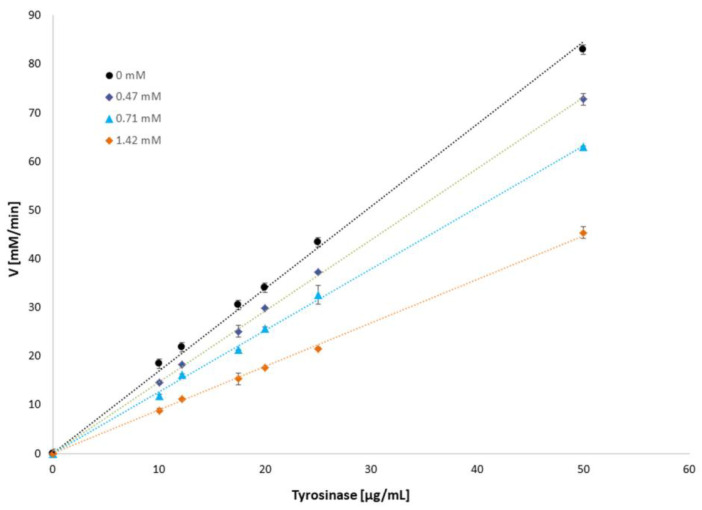
L-DOPA oxidation velocity [mM/min] dependence on tyrosinase concentration [µg/mL] in the presence of mangiferin (0 mM, 0.71 mM, 0.95 mM, 1.42 mM).

**Figure 5 antioxidants-12-01016-f005:**
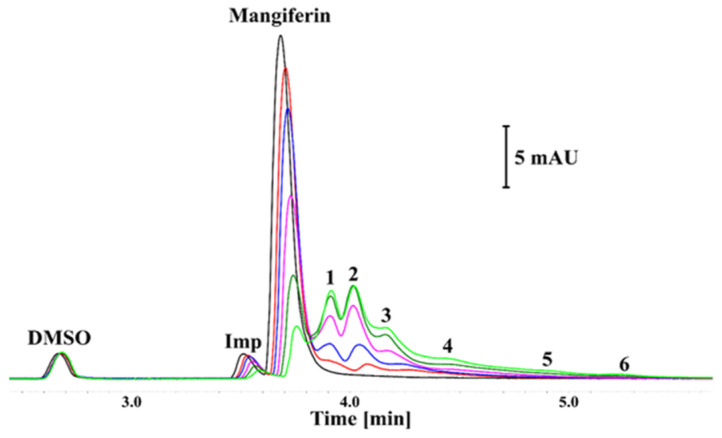
The CE analysis of (black trace) mangiferin solution in 50 mM phosphate buffer (pH 6.8) and its mixture with tyrosinase at the concentration of (red trace) 0.025 mg mL^−1^, (blue trace) 0.05 mg mL^−1^, (violet trace) 0.1 mg mL^−1^, (dark green) 0.15 mg mL^−1^, and (light green) 0.2 mg mL^−1^. The mangiferin concentration was constant in all solutions (0.3 mg mL^−1^). Imp—mangiferin impurity; the numbers indicate certain complexes of mangiferin with the enzyme.

**Figure 6 antioxidants-12-01016-f006:**
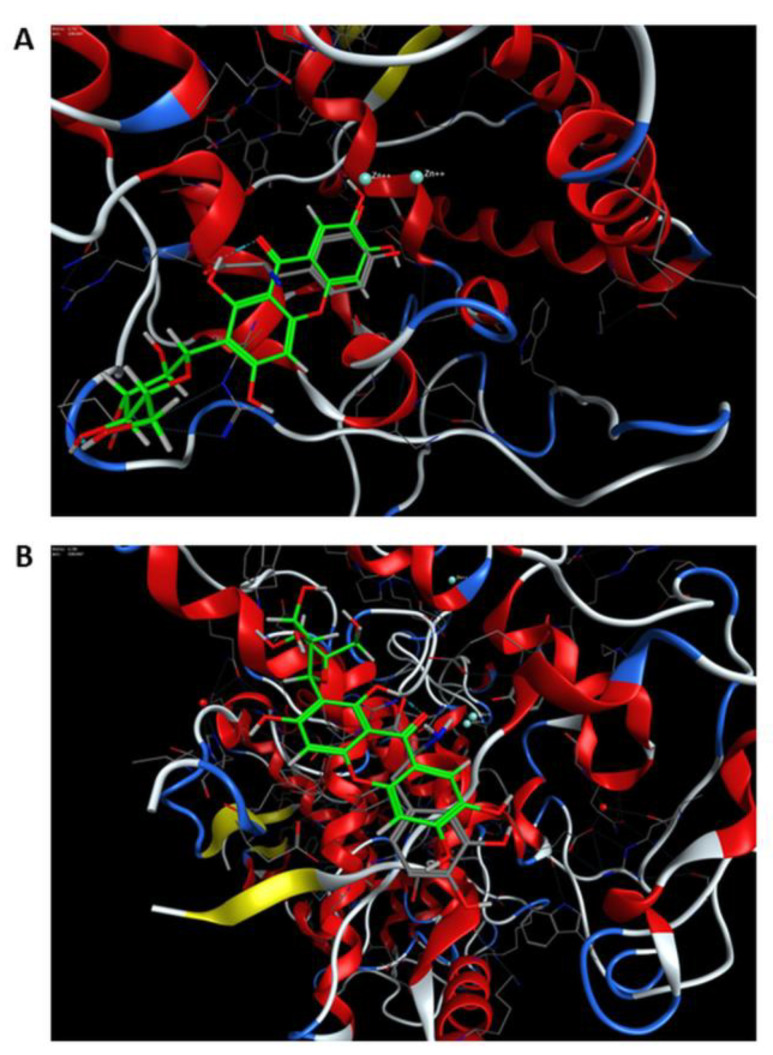
(**A**) The final pose of mangiferin (ligand highlighted in green) superposed with the original conformation of L−DOPA (ligand highlighted in gray) at the active site of tyrosinase (PDB ID: 4P6S). (**B**) The peripheral site of tyrosinase (PDB ID: 4P6S).

**Figure 7 antioxidants-12-01016-f007:**
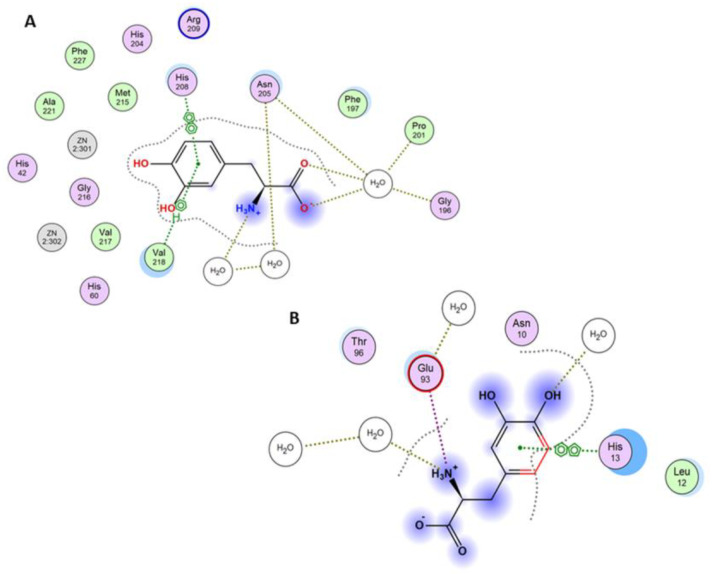
(**A**) Interaction diagram between the active site of tyrosinase and the original L−dopamine in the crystal structure. Legend: picture of two aromatic rings means a π-π interaction between the molecules, the H-ring describes a CH-π interaction, and the dotted green lines indicate a salt bridge. (**B**) Interaction diagram between the peripheral site of tyrosinase and the original L−dopamine in the crystal structure. Legend: picture of two aromatic rings means a π-π interaction between the molecules and the dotted green lines indicate a salt bridge. Names and numbers of protein amino acid residues interacting or surrounding the ligand are given in rings.

**Figure 8 antioxidants-12-01016-f008:**
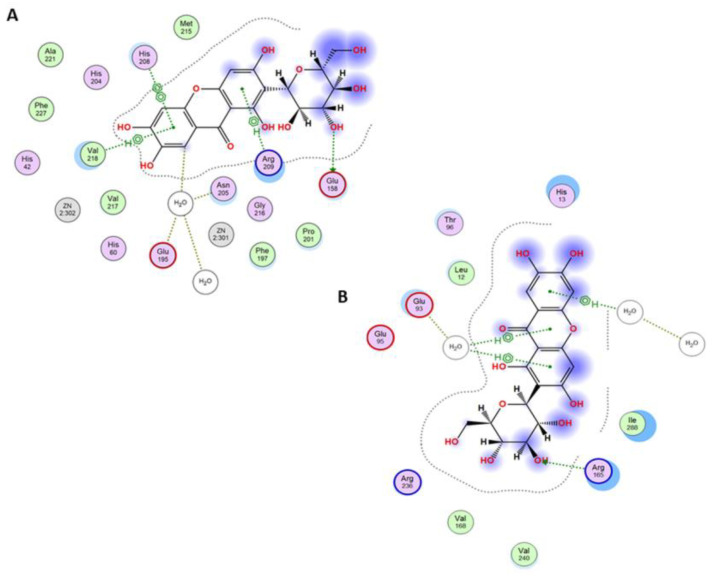
(**A**) Interaction diagram between the active site of tyrosinase and the mangiferin in the complex structure. Legend: picture of two aromatic rings means a π−-π interaction between the molecules, the H−ring describes a CH−π interaction, and the dotted green lines indicate a salt bridge. (**B**) Interaction diagram between the peripheral site of tyrosinase and the mangiferin in the complex structure. Legend: H−ring describes a CH−π interaction, and the dotted green lines indicate a salt bridge. Names and numbers of protein amino acid residues interacting or surrounding the ligand are given in rings.

## Data Availability

Samples of the compounds are available from the authors.

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
