# Peer review of "Mangiferin Affects Melanin Synthesis by an Influence on Tyrosinase: Inhibition, Mechanism of Action and Molecular Docking Studies"

_antioxidants, 2023, doi:10.3390/antiox12051016_

Round 1
Reviewer 1 Report
This study elucidates the enzymatic and molecular interaction between Mangiferin, which has been reported to have a correlation with various diseases and infections, and Tyrosinase. This is an interesting topic for readers in this research field. Typically, experimental results are accompanied by comparison controls such as positive and negative controls to confirm the validity of the results. However, a concern in this paper is that the effectiveness of Mangiferin cannot be accurately assessed due to the lack of positive controls shown in many experimental results. In Figure 2, the authors need to show the graph for the control used, kojic acid, similar to Mangiferin. Additionally, the presence of positive controls is essential in Figures 3-5, as this reviewer believes they will assist readers in comparing Mangiferin and kojic acid.
Minor,
Line 307–309:
“Additionally, between mangiferin molecule and protein an extra H-arene interaction is observed, with Arg209 and a salt bridge Glu158. Other interactions like salt bridge with Arg165 and more ligand-receptor exposures are observed (Figure. 8A).”
This reviewer was unable to understand the meaning of this text from Figure 8A. Please show Glu158 and Arg165 in the figure and revise the description to make it understandable for readers.
Typos,
Line 268
Figure X
Author Response
Dear Reviewer,
Thank you for the effort into evaluating our work, and for your valuable comments.
We included all suggestions and comments made by the three reviewers, and on this basis we decided to dispense with the interpretation of our results in relation to human skin
Please find enclosed responses to your comments.
This study elucidates the enzymatic and molecular interaction between Mangiferin, which has been reported to have a correlation with various diseases and infections, and Tyrosinase. This is an interesting topic for readers in this research field. Typically, experimental results are accompanied by comparison controls such as positive and negative controls to confirm the validity of the results. However, a concern in this paper is that the effectiveness of Mangiferin cannot be accurately assessed due to the lack of positive controls shown in many experimental results. In Figure 2, the authors need to show the graph for the control used, kojic acid, similar to Mangiferin. Additionally, the presence of positive controls is essential in Figures 3-5, as this reviewer believes they will assist readers in comparing Mangiferin and kojic acid.
The response
The experiments described in Section 3.1, which results were shown in Figures 2-4, were aimed to demonstrate the interaction of mangiferin and the tyrosinase enzyme.
The Figure 2 presented the influence of tyrosinase activity in the presence of mangiferin. According to the reviewer's recommendations, a positive control - kojic acid has been added. We agree, that the course of inhibition of positive control on the graph increases the possibility of assessing the effect of mangiferin on tyrosinase activity.
The Figure 3 and 4 have been created to estimate the direct influence of mangiferin on the tyrosinase activity, the type of inhibition (Figure 3) and reversibility of interaction tyrosinase – mangiferin (Figure 4). The analysis have been conducted without (0 mM) and with mangiferin (0.71 - 1.42 mM). In those two experiments positive control had not been done, cause in such analysis positive control is not provided. Mangiferin- tyrosinase and kojic acid- tyrosinase interactions have different nature and chemical structure and could not be summarized on one Figure. This would require additional analysis. Interactions among tyrosinase and kojic acid are widely described https://doi.org/10.3390/antiox11030502, https://doi.org/10.1111/j.2042-7158.1994.tb03253.x. Addition of positive control do not affect the interpretation of the results obtained and described on Figures 3 and 4, determining the interactions between mangiferin and tyrosinase.
The experiments described in Section 3.2, which results were shown in Figure 5, were aimed to demonstrate the interaction of mangiferin and the enzyme. The Affinity Capillary Electrophoresis (ACE) technique was used for this purpose. Several designs of ACE experiments have been described to date [Electrophoresis 2000, 21, 3905 - 3918]. The design used in our work was based on the shift of mobility of investigated compound - mangiferin [Ace. Chem. Res. 1995,28, 461-468]. This analyte feature characteristic UV spectrum which enables to confirm the identity of detected signal.
In the experiment, the ligand (mangiferin) solutions with and without the addition of the enzyme were analyzed. It was clearly presented that the mangiferin signal feature the shift of mobility when the enzyme is added. The change of signal area was proportional to the amount of added enzyme. The appearance of “additional signals” was in line with the molecular docking studies. These signals featured UV spectrum characteristic to mangiferin. We agree that in the experiment no positive control was used. However, it is not practiced in ACE. The interaction occurs or not. We agree that performance of similar experiment with known tyrosinase inhibitor (e.g. kojic acid) might contribute to the total value of the manuscript. However, such experiment would be a separate experiment that will provide additional knowledge but cannot act as positive control for the mangiferin-tyrosinase interaction assay. It is due to the different interaction of kojic acid with the enzyme.
Line 307–309:
“Additionally, between mangiferin molecule and protein an extra H-arene interaction is observed, with Arg209 and a salt bridge Glu158. Other interactions like salt bridge with Arg165 and more ligand-receptor exposures are observed (Figure. 8A).”
This reviewer was unable to understand the meaning of this text from Figure 8A. Please show Glu158 and Arg165 in the figure and revise the description to make it understandable for readers.
The response
The sentence was removed and has been replaced by more readable- lines: 291-295. The aminoacids residues in the Figure 8a and 8b are given in rings, the additional explanation of the Figures 7a and 7b -lines: 319-320; as well as Figure 8a and 8b had been added – lines: 327-328,
Typos, Line 268, Figure X
The response
In the line 254 Figure X was changed on Figure 5.
Reviewer 2 Report
The authors of the manuscript antioxidants-2293871 investigated the inhibition of tyrosinase by mangiferin. The manuscript is generally well-written. It needs minor corrections before publication.
The abstract must be reduced to 200 words.
The Introduction Section must be reduced in length. Some of the parts could be moved to the Discussion Section. Changes must be made in the first sentence, L46-L47, to make it more suitable for a scientific paper. I understand the marketing target - however, rural life also exposes to oxidative stress.
Section 2.2. Tyrosinase assay – I suggest changing it to “Tyrosinase inhibition assay.”
L150 - The kojic acid was used as a standard – I suggest completing “for the tyrosinase inhibition assay.”
The Antioxidants journal template must be followed – no empty spaces between paragraphs and/or Sub-Sections.
Author Response
Dear Reviewer,
Thank you for the effort into evaluating our work, and for your valuable comments.
We included all suggestions and comments made by the three reviewers, and on this basis we decided to dispense with the interpretation of our results in relation to human skin
Please find enclosed responses to your comments.
The authors of the manuscript antioxidants-2293871 investigated the inhibition of tyrosinase by mangiferin. The manuscript is generally well-written. It needs minor corrections before publication.
The abstract must be reduced to 200 words.
The response
The abstract was reduced in length. The lines 18-21, were appropriately changed.
The Introduction Section must be reduced in length. Some of the parts could be moved to the Discussion Section. I understand the marketing target - however, rural life also exposes to oxidative stress.
The response
According to the fact, that analysis had been conducted with the use of mushroom tyrosinase, not human the appropriate studies should be conducted. The authors decided to make changes in the manuscript. Sentences that refer to the possibility of using mangiferin as a tyrosinase inhibitor in the skin were removed from the introduction and description of the results (discussion), Removed part/parts are marked on red. Only a suggestion was left in the discussion, referring to the possibility of using mangiferin in mammals after further detailed studies using human tyrosinase and tissue cultures. Lines 405-411.
Changes must be made in the first sentence, L46-L47, to make it more suitable for a scientific paper.
The response
The sentence was removed from the manuscript.
Section 2.2. Tyrosinase assay – I suggest changing it to “Tyrosinase inhibition assay.”
The response
The Section 2.2 title in the line 124 was supplemented with the text suggested by the Reviewer
L150 - The kojic acid was used as a standard – I suggest completing “for the tyrosinase inhibition assay.”
The response
The sentence in the line 144 was supplemented with the text suggested by the Reviewer
The Antioxidants journal template must be followed – no empty spaces between paragraphs and/or Sub-Sections.
The response
The Antioxidants journal template had been followed.
Reviewer 3 Report
The topic is of interest; however, the manuscript requires significant revisions prior considerations.
Based on the description of the overview of the problem, rather random citations for some aspects of studies, it is concluded that the authors have gaps in understanding of tyrosinase functions and melanogenesis. For example, citation of Jiratchayamaethasakul et. all. 2020 [38] for tyrosinase assay documents significant gap in this area. Cite older more representative papers on measurement of DOPA oxidase activity of tyrosinase.
The description of the assay is not precise. Usually, 1 mM L-DOPA is used as the substrate and dopachrome formation is measured at 475 nm. This will allow to calculate umols of dopachrome formed using cuvette with 1 cm light path. Dopachrome coefficient is 3600.
In this context: "The kojic acid was used as a standard. According to the data supplier: 1U of tyrosinase hydrolyses 1 µM L-DOPA/min at the experimental conditions. The L-DOPA molar 151
absorption coefficient is 3800 M–1 cm–1" does not make sense. Tyrosinase does not hydrolyze L-DOPA!!!!
Why mushroom tyrosinase is used by not mammalian. The latter could be extracted easily from the pigmented melanomas or melanocytes as it was done in the past.
Why crystal structure of tyrosinase from Bacillus megaterium with L-DOPA in the active site is used for docking experiments, If the authors want to refer their findings to human pigmentation, human tyrosinase for docking experiments should be used.
line 192: "s removing the ligand (L-dopamine)?????
While pigment cells in culture were not used to study inhibition of melanogenesis.
Introduction and discussion are disconnected from the real data and methodology presented.
The manuscript is full of errors starting from the abstract "diphenolase activity of ty-rosinase with L-dopamine (L-DOPA) as a substrate."
Author Response
Dear Reviewer,
Thank you for the effort into evaluating our work, and for your valuable comments.
We included all suggestions and comments made by the three reviewers, and on this basis we decided to dispense with the interpretation of our results in relation to human skin
Please find enclosed responses to your comments.
The topic is of interest; however, the manuscript requires significant revisions prior considerations.
Based on the description of the overview of the problem, rather random citations for some aspects of studies, it is concluded that the authors have gaps in understanding of tyrosinase functions and melanogenesis. For example, citation of Jiratchayamaethasakul et. all. 2020 [38] for tyrosinase assay documents significant gap in this area. Cite older more representative papers on measurement of DOPA oxidase activity of tyrosinase.
The response
The citations have been changed on: doi: 10.1023/a:1023620501702. PMID: 12765534, doi: 10.1021/jf011378z. PMID: 12083892, that the authors consider more precisely describing the method we use in our analyses. The additional information concerning methodology were added in lines 125-133
The description of the assay is not precise. Usually, 1 mM L-DOPA is used as the substrate and dopachrome formation is measured at 475 nm. This will allow to calculate umols of dopachrome formed using cuvette with 1 cm light path. Dopachrome coefficient is 3600.
In this context: "The kojic acid was used as a standard. According to the data supplier: 1U of tyrosinase hydrolyses 1 µM L-DOPA/min at the experimental conditions. The L-DOPA molar 151
absorption coefficient is 3800 M–1 cm–1" does not make sense. Tyrosinase does not hydrolyze L-DOPA!!!!
The response
Following the recommendation of the reviewer, the above sentence has been removed from the text of the manuscript. We agree with the reviewer: tyrosinase does not hydrolyze L-DOPA, this statement has been corrected.
The L-DOPA oxidation product – dopachrome formation according to the literature is 475nm, the analysis was performed on 480nm, according to maximum wavelength of dopachrome formation, determinated in our laboratory conditions.
Why mushroom tyrosinase is used by not mammalian. The latter could be extracted easily from the pigmented melanomas or melanocytes as it was done in the past.
While pigment cells in culture were not used to study inhibition of melanogenesis.
Introduction and discussion are disconnected from the real data and methodology presented.
The response
During the course of experiments neither human tyrosinase nor adequate cell cultures had not been available. Therefore, the analysis had been conducted with the use of mushroom tyrosinase. However, we agree with the Reviewer that for the application to humans, appropriate studies should be conducted. So, as suggested by the Reviewer, sentences that refer to the possibility of using mangiferin as a tyrosinase inhibitor in the skin were removed from the introduction and description of the results (discussion), Removed part/parts are marked on red. Only a suggestion was left in the discussion, referring to the possibility of using mangiferin in mammals after further detailed studies using human tyrosinase and tissue cultures. Lines 405-411.
Why crystal structure of tyrosinase from Bacillus megaterium with L-DOPA in the active site is used for docking experiments, If the authors want to refer their findings to human pigmentation, human tyrosinase for docking experiments should be used.
The response
Concerning crystal structure we have respond what X-ray structures are available and why we have used this particular one. The crystal structure of tyrosinase from Bacillus megaterium was the only one with ligands (L-DOPA) in two different positions. Taking any other structure without ligands is less accurate in docking because we applied so called bound docking and not unbound docking. The first type of docking is made using structure of the target the same as in the complex. In the second one with apo version of the protein it does not consider potential induce fit of the target after binding.
For the further studies the authors will use the human tyrosinase for docking experiments.
line 192: "s removing the ligand (L-dopamine)?????
The response
The sentence was changed, Line 181
The manuscript is full of errors starting from the abstract "diphenolase activity of ty-rosinase with L-dopamine (L-DOPA) as a substrate."
The response
Following the recommendation of the reviewer, the above sentence has been removed from the text of the manuscript.
Round 2
Reviewer 1 Report
The authors have addressed the reviewer's concerns.
A minor spell check is needed.
Line 157
“leaving the recognized ligands binding site (L-DOPA) empty..”
Line 268
Ffurther
Author Response
April 07, 2023
Dear Reviewer,
Thank you for the effort into evaluating our work, and for your valuable comments.
We included all spelling suggestions in the manuscript:
Please find enclosed corrections to your comments.
Line 157: “leaving the recognized ligands binding site (L-DOPA) empty..”
The response:
Line 168: leaving the recognized ligands binding site (L-DOPA) empty.
Line 268: Ffurther
The response:
Line 284: was corrected: furthermore
Kind regards
The authors of the manuscript
Reviewer 3 Report
The authors effort to correct the manuscript is appreciated, which included removal of embarrassing errors that started from the abstract. However, many deficiencies remain and some are critical to correct.
As relates to the abstract it is corrected in the pdf file but not the abstract on line, which indicate lack of attention to detail from the senior author.
This lack of attention to the detail is further amplified in the section describing tyrosinase assay. You cite both by author name and by number the reference to mushroom tyrosinase. Better older referral for such assay could be used since researchers are measuring catecholase activity of mushroom tyrosinase for almost 100 years.
Important, crucial details of the assay are not included, such as concentration of the substrate. Also, the English is awkward (The product formation of L-DOPA oxidation), say dopachrome formation. Use of 480 nm but not 475 nm as everybody in the field was using for almost 50 years, would prevent from calculation of product formation in umols, e.g., É› = 3600 M- 1 cm- 1 (molar absorption coefficient for the product dopachrome) at 475 nm.
Since authors discuss the implications of the findings in the context of mammalian system, validation of the results using mammalian tyrosinase is required. B16 melanoma, available almost in any laboratory working on pigmentation, if human line is not available, is sufficient. The experiments must be performed. You can use extracts or partially purify the enzyme. Assays would include tyrosine hydroxylase activity and dopa oxidase activity which are very easy and depending on the preference could be colorimetric, radiometric or fluorometric ((J Cell Sci 89, 287-296, 1988; Int J Biochem 16, 323-326, 1984; Bioscience Reports 4, 1059-1064; 1984).
As relates to docking analyses, the authors had no choice since crystal structure of human tyrosinase is not available. However, the crystal structure of TRP1 is available, and based on homology, they could generate good models of human tyrosinase and perform molecular modeling. The above are necessary to relate experimental findings to mammalian pigmentation. The significance of tyrosinase inhibitors would be of importance also under pathological conditions (Frontiers in Oncology 2022;12. DOI: 10.3389/fonc.2022.842496)
Finally, please ask someone proficient in scientific English to correct the manuscript and pay attention to details.
Author Response
April 07, 2023
Dear Reviewer,
Thank you for the effort into evaluating our work, and for your valuable comments.
We included all suggestions and comments made by the three reviewers, and on this basis we decided to dispense with the interpretation of our results in relation to mammals.
Please find enclosed responses to your comments:
As relates to the abstract it is corrected in the pdf file but not the abstract on line, which indicate lack of attention to detail from the senior author.
The response
When adding a revised manuscript in response to reviews, there is no option to reload only the abstract that is seen on line. Accordingly, we couldn't change the on line text of abstract added during the first submission.
This lack of attention to the detail is further amplified in the section describing tyrosinase assay. You cite both by author name and by number the reference to mushroom tyrosinase. Better older referral for such assay could be used since researchers are measuring catecholase activity of mushroom tyrosinase for almost 100 years.
The response
The authors cited the manuscripts in which the mushroom tyrosinase was used, because the one was used in conducted studies. The citation 38 was changed on :
Yagi A, Kanbara T, Morinobu N. Inhibition of mushroom-tyrosinase by aloe extract. Planta Med. 1987 Dec;53(6):515-7. doi: 10.1055/s-2006-962798. PMID: 17269093.
Important, crucial details of the assay are not included, such as concentration of the substrate. Also, the English is awkward (The product formation of L-DOPA oxidation), say dopachrome formation. Use of 480 nm but not 475 nm as everybody in the field was using for almost 50 years, would prevent from calculation of product formation in umols, e.g., É› = 3600 M- 1 cm- 1 (molar absorption coefficient for the product dopachrome) at 475 nm.
The response
The concentration of L-DOPA was added in line: 125
“The product formation of L-DOPA oxidation” was changed to “dopachrome formation”, line 115
The dopachrome formation according to the literature is 475nm, the analysis was performed on 480nm ± 6nm, according to maximum wavelength of dopachrome formation, determinated in our laboratory conditions. Because the maximum absorbance of the dopachrome is within the standard deviation we can write 475 nm: 480 nm was changed to 475 nm in the line 126
Since authors discuss the implications of the findings in the context of mammalian system, validation of the results using mammalian tyrosinase is required. B16 melanoma, available almost in any laboratory working on pigmentation, if human line is not available, is sufficient. The experiments must be performed. You can use extracts or partially purify the enzyme. Assays would include tyrosine hydroxylase activity and dopa oxidase activity which are very easy and depending on the preference could be colorimetric, radiometric or fluorometric ((J Cell Sci 89, 287-296, 1988; Int J Biochem 16, 323-326, 1984; Bioscience Reports 4, 1059-1064; 1984).
The response
We kindly thank the reviewer for pointing out the publications and methods related to human tyrosinase, however our manuscript in the current form concerns only mushroom tyrosinase and the relation to the food.
We agree with the reviewer that the indicated methods are necessary for research on pigmentation changes in humans. This was briefly highlighted in the Discussion section, lines: 407-415.
As relates to docking analyses, the authors had no choice since crystal structure of human tyrosinase is not available. However, the crystal structure of TRP1 is available, and based on homology, they could generate good models of human tyrosinase and perform molecular modeling. The above are necessary to relate experimental findings to mammalian pigmentation. The significance of tyrosinase inhibitors would be of importance also under pathological conditions (Frontiers in Oncology 2022;12. DOI: 10.3389/fonc.2022.842496)
The response
The authors are grateful for finding a suitable, most human-like tyrosinase model for use in the docking studies. However, in the current form of the manuscript, we do not relate our own results done on mushroom tyrosinase to the mammals. The significance of tyrosinase inhibitors in pathological conditions in mammals is important, but not in the present form of the manuscript. More detailed research in this area will be conducted in the future. For those reasons the last lines of the Discussion section were changed, lines 411-415.
Finally, please ask someone proficient in scientific English to correct the manuscript and pay attention to details.
The response
Language proofreading was done by a professional company. Changes made to the text are shown in the track changes option. The translation certificate was sent to the Editor.
Kind regards
The authors of the manuscript
Round 3
Reviewer 3 Report
The authors instead of performing confirmatory experiments using mammalian system decided to discuss with the editor.
They say "{ However, in the current form of the manuscript, we do not relate our own results done on mushroom tyrosinase to the mammals. The significance of tyrosinase inhibitors in pathological conditions in mammals is important, but not in the present form of the manuscript. More detailed research in this area will be conducted in the future", which is not fully correct since they relate their findings also to human and animal systems.
With refocus strictly on plants, the manuscript would be of lower importance for the antioxidants and more appropriate for journals that deal with plants, mushrooms or nutrition.
Author Response
April 24, 2023
Dear Reviewer,
Thank you for the effort into evaluating our work, and for your valuable comments.
We included all suggestions and comments made by the three reviewers and Academic Editors. On this basis we decided to dispense with the interpretation of our results in relation to mammals and refer only to the browning process of food products. Thank you for all your suggestions and comments regarding the use of human tyrosinase and cell line studies. We will use them in further work.
Kind regards
The authors of the manuscript
